# Analysis of the Acoustic Comfort in Hospital: The Case of Maternity Rooms

**Simone Secchi** [1,*] **, Nicoletta Setola** [1,2] **, Luca Marzi** [1,2] **and Veronica Amodeo** [1]

1   Department of Architecture, University of Florence, Via San Niccolò 93, 50125 Florence, Italy;
    nicoletta.setola@unifi.it (N.S.); luca.marzi@unifi.it (L.M.); veronica.amodeo@unifi.it (V.A.)
2   Inter-University Research Center TESIS "Systems and Technologies for Social, Health and Education
    Structures", Via San Niccolò 93, 50125 Florence, Italy
*   Correspondence: simone.secchi@unifi.it

**Abstract:** Hospitals include a variety of different spaces with different requirements and levels of sensitivity to noise but also different activities and equipment that can cause high noise levels. In this article, noise disturbance in hospitals is studied with reference to a case study in a maternity ward. The analysis is carried out by means of sound insulation measurements between bedrooms and between bedroom and corridor. Equivalent sound pressure level measurements were carried out continuously for two days and nights. The number of awakening events is examined for each hour of the two nights. In addition, the results of a questionnaire conducted on more than 100 patients are reported. The results of the study show that the main cause of noise disturbance is activity in the corridors and that this kind of disturbance is usually repeated throughout the night. This is made more critical by the poor acoustic performance of the doors, but also by the habit of keeping doors open or half-open to allow doctors to always control patients. The article proposes some possible solutions to reduce noise intrusion from the corridor to the rooms.

**Keywords:** noise disturbance; hospitals; sound insulation; awakenings



## 1. Introduction

This paper is an extended version of the paper presented at the Inter Noise Conference 2019 [1].

There are many causes of stress in the hospital environment that can adversely affect the psycho-physical well-being of patients, visitors, and healthcare staff [2]; some of these environmental stressors include noise pollution, inadequate lighting, unpleasant odors, or too-high temperature [3].

Noise pollution is considered one of the most disturbing factors within hospital environments [4,5]. For healthcare staff, for example, prolonged exposure to noise can alter stress levels, reduce work performance [6,7] and increase burnout rates [8].

The sick patient is the most vulnerable individual within care spaces, being more sensitive to noise than a healthy individual [9,10]; furthermore, excessive noise level often negatively affects the patient's well-being and recovery process [11]. According to the World Health Organization (WHO), noise disturbance within hospital wards is directly related to the well-being and psychophysical response of patients, affecting the quality of rest, cardiovascular response, and the healing process [4].

Hospitals encompass a variety of different spaces, each of which has specific requirements, in relation to the activities that are carried out, the equipment present, and based on the level of sensitivity of the users when they are inside them.

Scientific laboratory studies conducted on women show that noise from medical devices, technical equipment, and alarms within a typical critical care unit (CCU) negatively affect the quality of physiological sleep [12]. Some surveys, carried out on 203 patients (121 males and 82 females) of intensive care units (ICU), highlight that interruptions caused

by human interventions, diagnostic testing, and alarms are considered the most disturbing source of noise during the sleeping hours [13].

Studies on the effects of noise on sleep, which have taken into account variables such as the variation of the duration and depth of the sleep phases by electroencephalography (EEG) [14], have shown that an equivalent sound level $L_{A,eq}$ of 45–50 dB(A) can alter the EEG tracing for about 50% of the exposed subjects. As a consequence, the day after a night of noise disturbance, sleep deprivation effects, such as a drop in alertness and momentary onset of light sleep (microsleep), can occur [15]. Similar values are found when considering environments dedicated exclusively to children or infants, such as the NICU (neonatal intensive care unit).

In maternity wards, excessive noise levels have adverse effects on pregnant or postpartum women, especially those with high-risk pregnancies, who may suffer from sympathetic nerve vasoconstriction, increased blood sugar, or increased cardiac frequency [16,17]. Pregnant women and elderly are considered more sensitive to noise due to more fragmented sleep patterns. Several studies, carried out from 2006 to 2010, highlighted a relationship between long-term noise and the risk of premature birth [18]; similarly, inadequate natural or artificial lighting affects the regular perception of the circadian rhythms of mother and newborns [19,20]. Moreover, noise affects mothers much more on the postnatal ward than on the labor ward [21].

The World Health Organization points out that during the night it is advisable to have an indoor equivalent sound pressure level ($L_{Aeq}$) not greater than 30 dB(A) (averaged over 8 h), and a maximum sound pressure level ($L_{Amax}$) not greater than 45 dB(A) for non-continuous noise. The Recommended Standard for Newborn Intensive Care Unit Design states that in any inpatient or newborn care area the combination of continuous background noise and transient noise, i.e., noise generated by staff and equipment, should not exceed an hourly $L_{Aeq}$ of 50 dB(A) [3]. Despite this, many international studies have monitored sound pressure levels within hospital environments that exceed this recommended value, with night-time $L_{Aeq}$ values between 38.7 and 68.6 dB(A) [22], or between 38 and 57 dB(A) with maximum peaks of 116 dB(A) [23].

In line with the scientific studies showing negative relationships between hospital noise and sleep quality [12–15], in this article we focus on patients' perception of noise inside hospital inpatient rooms during nighttime hours.

This work was carried out at the maternity ward of the Azienda Ospedaliero Universitaria of Careggi, Florence, Italy (A.O.U.C.), in the obstetrics ward B on the second floor of pavilion 7. The primary objective was to verify the environmental discomfort in a typical hospital ward bedroom configuration through:

- measurements of sound insulation between adjacent bedrooms and between bedrooms and corridor;
- continuous long-term measurements of sound pressure level inside the bedroom due to the equipment and to the activities in the corridor.

In addition, the results of previously presented questionnaires [1], distributed to the patients and dealing with the perception of acoustic comfort, are recalled to confirm the causes of acoustic discomfort.

## 2. The Acoustic Requirements for Hospital Bedrooms

In each country, there are different descriptors and related limit values for acoustic requirements of hospitals. Some countries deal with only the requirements of sound insulation between bedrooms (vertical and/or horizontal partitions) while other countries give recommendations or mandatory limit values also for the partition between the corridor and the bedroom. A detailed comparison between descriptors and limit values used in different European countries is reported in Refs. [24,25].

This study refers to an Italian hospital as a case study; therefore, a detailed description of Italian requirements for hospital bedrooms is reported.

The main Italian legislative references for the acoustic requirements of hospitals are the Decree of 5 December 1997 [26] about "Passive acoustic requirements of buildings", which is applicable to all (public and private) hospitals built after 1997, and the Decree of 11 October 2017 [27] about "Minimum Environmental Criteria", applicable only to public hospitals built or refurbished after 2017.

The limit values reported in the Decree of October 2017 for public hospitals refer to the Italian standards UNI 11367:2010 [28], for insulation requirements (sound insulation, impact noise and noise from equipment), and to UNI 11532:2018 [29], for the room acoustic requirements (Reverberation Time and Speech Transmission Index). The limit values for sound insulation performance are given by the Annex A (with reference to the superior performance) and by Annex B (with reference to the good performance) of UNI 11367:2010, except for the limit value of sound insulation of façades ($D_{2m,nT,w}$) which refers to the most restrictive standards of the Decree of 5 December 1997 [26].

Table 1 shows the limit values for airborne sound insulation, impact noise, and noise from service equipment, for Italian hospitals.

**Table 1.** Limit values of airborne sound insulation, impact noise, and service equipment noise for public hospitals in Italy.

| Descriptor | Limit value |
|---|---|
| Façade sound insulation | $D_{2m,nT,w} \geq 45$ dB * |
| Vertical partition between two hospital rooms (without doors) | $D_{nT,w} \geq 50$ dB ** |
| Vertical partition between corridor and room (with door) | $D_{nT,w} \geq 30$ dB ** |
| Horizontal partition between two hospital rooms | $D_{nT,w} \geq 55$ dB ** |
| Partition (vertical or horizontal) between hospital and other properties | $R'_w \geq 56$ dB ** |
| Horizontal partition between two hospital rooms | $L'_{n,w} \leq 53$ dB ** |
| Partition between hospital and other properties | $L'_{n,w} \leq 58$ dB ** |
| Equivalent sound pressure level from service equipment (continuous operation) | $L_{eq} \leq 25$ dB(A) * |
| Corrected maximum sound pressure level from service equipment (discontinuous operation) | $L_{id} \leq 34$ dB(A) ** |

(*) Value referred to the Decree of 5 December 1997. (**) Value referred to the Decree of 11 October 2017.

The descriptor used for the airborne sound insulation between rooms inside the hospital is $D_{nT,w}$ and not $R'_w$, because sound transmission between two adjacent rooms is due both to the direct sound path through the partition wall and to the indirect sound paths through the air ducts or the corridor.

The corrected equivalent sound pressure level, $L_{id}$, from service equipment with discontinuous operation is calculated as:

$$L_{id} = L_{ASmax} + K_2, \tag{1}$$

where:

$$K_2 = -10 \log (T/T_0) \tag{2}$$

T = average reverberation time in the room (s);
$T_0$ = reference reverberation time:
for $V \leq 100$ m$^3$, $T_0 = 0.5$ (s);
for $100 < V < 2500$ m$^3$, $T_0 = 0.05$ (V)$^{0.5}$ (s);
for $V \geq 2500$ m$^3$, $T_0 = 2.5$ (s).

Regarding the requirement of reverberation time, the decree of 11 October 2017 refers to the first part of UNI 11532:2018, where general procedures and the definition of descriptors are covered; at the moment, the specific part of the standard, which will contain limit values for hospitals, has not been published.

## 3. Materials and Methods

### 3.1. Description of the Case Study

The new maternity ward examined is part of the Careggi General Hospital, consisting of more than 50 buildings spread over more than 200,000 square meters. The analyzed building is the result of a pre-existing building partial renovation and consists of 5 floors connected to the block of the so-called free professions containing outpatient activities. The new maternity ward contains outpatient activities on the basement and ground floors and inpatient activities on the upper floors. The research was developed specifically by analyzing the rooms belonging to the inpatient wards.

The wards are organized according to a five-fold structure, characterized by a twin-aisle spatial layout; the services are located in the central area and the rooms on the windowed sides of the building (Figure 1a). The maternity ward is equipped with two-bed coupled rooms, which follow the same module; each room (floor area and volume 23 m$^2$ and 68 m$^3$, respectively) has its own toilet directly accessible from the room (Figure 1b) and overlooks the internal corridors that connect the ward to the hall distribution system. The distribution system does not provide for a filtering space between the rooms and the corridor; in fact, as it is usual in a hospital ward, the patient rooms are in close contact with all the flows of people and correlated activities that reach the ward daily, without any filters able to "isolate" the in-patient from the routine activities of the health services.

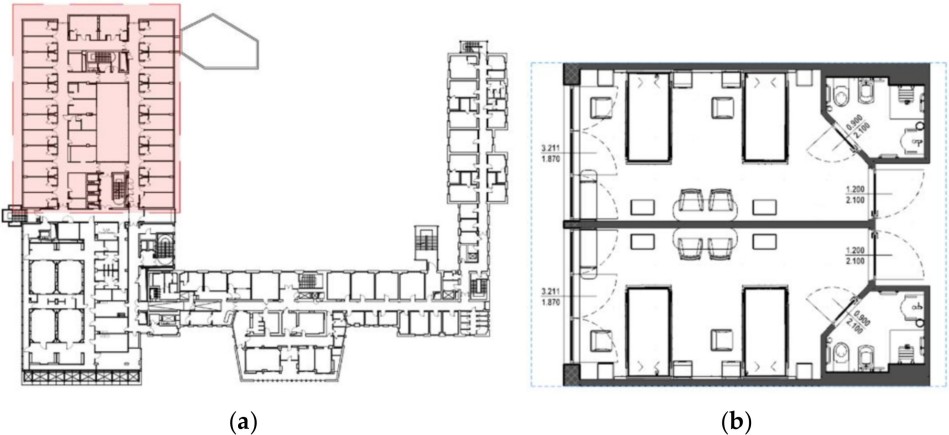

(a)                                                                (b)

**Figure 1.** Careggi General Hospital of Firenze; (**a**) on the left side: the maternity ward layout (marked in red); (**b**) on the right side: a typical two-bed coupled rooms module.

Room layout is typical of a two-bed inpatient unit with an indoor toilet, without the "family zone." The room is divided into a distribution area, in front of the access to the toilet, and a pair of beds with the bed-head on the same wall, while the windowed wall is on the opposite side of the toilet. The dimension of the bedroom and the distributional system of the maternity ward of the Careggi Hospital is typical of many other hospitals realized all over the world.

The building, completed in 2010, has a reinforced concrete structural skeleton, with the partition walls made of a double sheet of plasterboard on both sides. The wall central cavity, used as electrical, medical gas, and data network systems passage, is filled with rock wool, while toilet drains are housed in an internal cavity shared by the two toilets.

Figure 1b shows the bedrooms analyzed with acoustic measurements.

The door, in Figures 2 and 3, between the room and the corridor, is a roto-translating door with no bottom rebate; for this reason, this is the weak element in terms of sound transmission. Figure 2 shows details of the junctions between the door and the lateral wall and between the door and the flooring. Large gaps are evident on both joints.

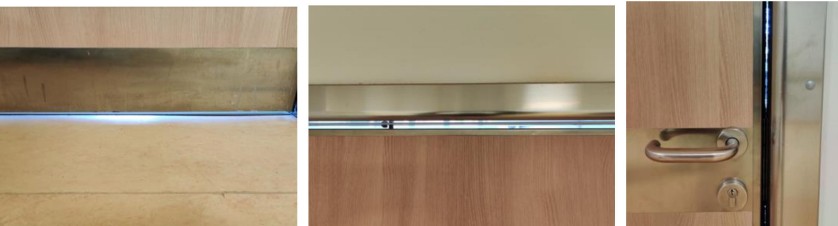

**Figure 2.** Details of the roto-translating door between the room and the corridor with no bottom rebate.

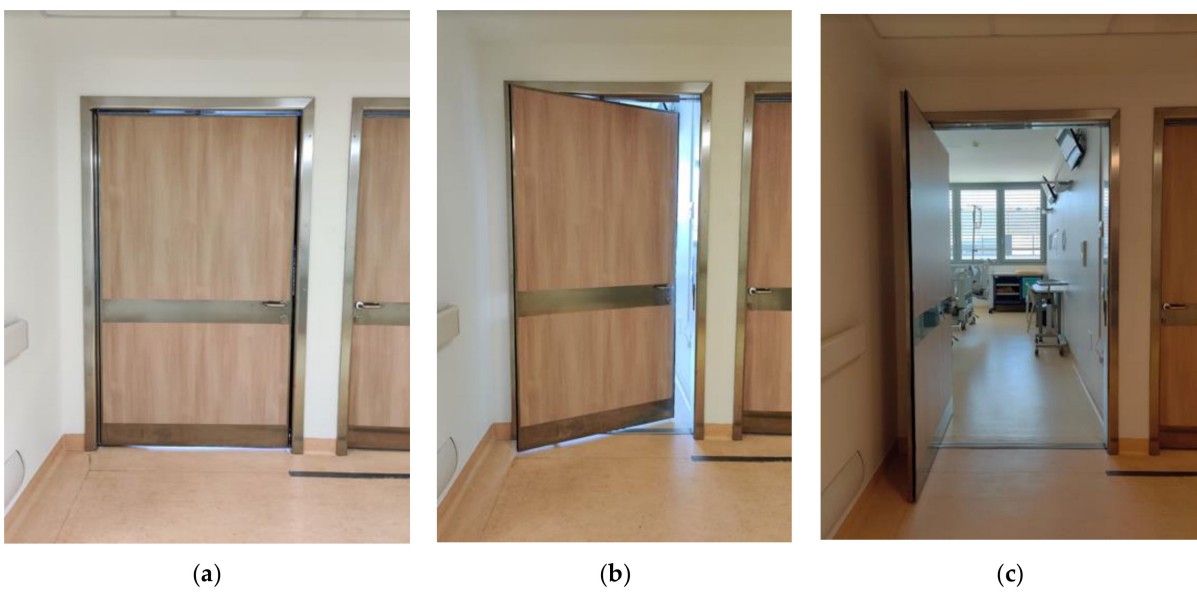

| (a) | (b) | (c) |

**Figure 3.** The three configurations of the door: (**a**) closed, (**b**) half-open, and (**c**) open.

In a contemporary hospital system, the use of digital systems for the management of information flows, involving heterogeneous and multi-disciplinary data and knowledge, is now well established. When dealing with issues related to Healthcare Facility Management and Healthcare Space Management activities, the systems defined as CAFM (Computer-Aided Facility Management), integrated with BIM (Building Information Modeling) systems, are strategic in order to make information and interlocutors communicate with each other, extrapolating those elements that characterise the functioning of the management process iter [30].

In the case of the Careggi polyclinic, the healthcare company uses a digital suite that integrates BIM–CAFM and GIS systems, giving back information on web applications accessible to all the company's referents. The system, called SACS© (acronym for structural consistency analysis), correlates structural (shape geometries and dimensions of buildings and rooms), technological (types of electrical, mechanical, data networks and electro-medical equipment), and organisational information (types of healthcare environments, types of care, identification and classification codes, personnel, etc.).

The BIM models of the buildings inter-operate with the information of SACS, in order to perform analysis (simulations) related to specific environmental verifications, for example, those related to energy performance or to the air-lighting ratios of the rooms. In order to integrate these data also with the information on acoustic comfort in the hospital bedrooms, sound propagation simulations, based on the information contained in SACS, have been carried out. Unfortunately, at the moment, the software used for the acoustic simulations (Ramsete®) cannot dialogue directly with BIM systems (it is not able to manage the data import-export criteria according to the IFC standards defined by Building Smart

International), but the results of the simulations have been catalogued so that they can be included in the BIM models as attributes related to environmental units (bedrooms).

### 3.2. Acoustic Measurements Methodology and Data Analysis

Sound insulation measurements have been carried out according to the standard ISO 16283-1 [31]. Standardized level difference, $D_{nT}$, and normalized level difference, $D_n$, have been measured between corridor and bedroom and between adjoining bedrooms. In case of the adjoining bedrooms partition, also apparent sound reduction index, R', has been measured. The ratings of the above-mentioned descriptors have been calculated according to the standard ISO 717-1 [32].

The measurement instrumentation used consisted of a two-channel class 1 real-time analyzer, model 01 dB Symphonie, equipped with preamplifiers and $\frac{1}{2}$ inch diffuse field Gross microphones. Before each measurement, the microphones were calibrated with a signal of 94 dB at 1000 Hz with a 01dB calibrator.

A single dodecahedron omnidirectional loudspeaker, model 01 dB LS1, and five microphone positions in both the source and receiving rooms were placed (Figure 4, left).

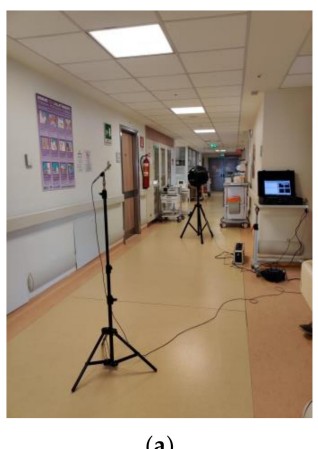
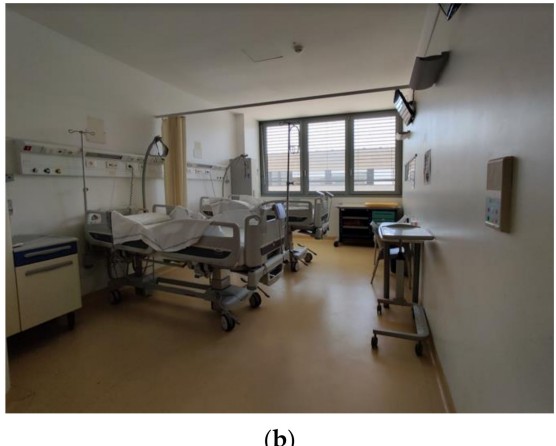

(**a**)                                          (**b**)

**Figure 4.** Careggi General Hospital of Firenze during measurements: (**a**) on the left side the corridor and (**b**) on the right side the bedroom.

A Maximum Length Sequence (MLS) technique [33] was used for the generation and the analysis of the sound level difference. This technique allowed us to carry out measurements during normal corridor use, being independent of background noise.

Figure 4 shows part of the instrumentation used to measure the sound insulation between bedrooms.

The sound transmission between corridor and bedroom is generally dominated mainly by the door performance. In some cases, the transmission between two adjoining bedrooms can also be greatly affected by the transmission through the corridor, as described by EN ISO 12354-1, annex H [34]. For this reason, measurements of sound insulation between corridor and bedroom have been carried out in three different conditions:

- With the door closed, Figure 3a;
- With the door half-open (open by 20 cm), Figure 3b;
- With the door completely open, Figure 3c.

The condition with the door half-open has been considered since this is the most used condition in hospital wards as it allows healthcare staff to maintain an auditory and, in some cases, visual contact with patients.

Continuous Sound Pressure Level measurements were carried out with a class 1 real-time analyzer (same instrumentation used for sound insulation measurements, see above) from 11:30 a.m. on Friday to approximately 10:30 p.m. on Sunday, inside a bedroom that

was not used during all the measurement period, so that only noise from outside the room (occupied adjoining bedroom or corridor) was considered.

The time history of the complete measurement (with recording sampling every 2 s) was analyzed, highlighting the peaks with a sound level capable of causing sleep deprivation in the patient.

According to the studies on the effects of noise on sleep reported in the introduction [13] and with the contents of the WHO Guidelines [4], the threshold for detecting events capable of causing awakening was set at 45 dB(A).

The over-threshold ($L_{Aeq}$ = 45 dB(A)) events were selected via a trigger set in the 01dBtrait® 5.2 software, a tool distributed by Aesse Ambiente srl (Trezzate, Italy); then events were counted for each hour of the two examined nights.

### 3.3. Questionnaires

Questionnaires were administered to a sample of 108 people (about 70% patients and 30% companions) split into 26–35 years range for 51% of the cases and between 36–45 years for 37% of cases, with a stay in the ward of typically 2–3 days (57% of the patients interviewed, while 29% stayed 4–6 days and only in a few cases stayed a week or more).

The detailed survey description, which was composed of 8 questions referring to the acoustic comfort perceived by patients inside the ward, was presented at the Inter Noise Conference 2019 [1]. In this paper, we refer only to the main results of this analysis.

Questions (How often did you wake up at night due to noise? After waking up, can you go back to sleep easily? How often are you disturbed during the day by sounds? How much do you think the area near the patient rooms is silent during the night hours? Have you heard conversations coming from the hallway? Which are the typical sources of noise?) were aimed at estimating the acoustic quality of the nighttime hours within the department and what the most disturbing noise sources were.

### 3.4. Acoustic Simulations Methodology

To study the distribution of sound pressure level in the typical two-bed coupled rooms, a 3D modeling of the bedrooms was realized and simulations were carried out using Ramsete® 2.7, a software distributed by Spectra srl (Vimercate, Italy), able to replace the sound rays, typical of ray tracing, with divergent pyramidal beams whose axes coincide with the original rays [35]. The generation of pyramids is perfectly isotropic, through progressive subdivision based on the powers of number 2 (subdivision level: $8 \times 2^n$, with minimum subdivision level value set to 8). The accuracy of the sound propagation software model in enclosed spaces is presented in literature [35].

In the model, the partitions between source and receivers were considered obstructing, with frequency values of the Sound Reduction Index taken from the database of the software and corresponding to the kind of partitions (walls and door) described in the hospital BIM model. Since the transmission from the corridor to the bedroom is due mainly to the door, Table 2 shows only the values of the Sound Reduction Index and of the absorption coefficient of the door, which correspond to a typical hospital door made of a sandwich panel with laminate skins, without seals on the side joints.

**Table 2.** Sound Reduction Index and absorption coefficient of the door between bedroom and corridor.

| Frequency (Hz) | 125 | 250 | 500 | 1000 | 2000 | 4000 |
|---|---|---|---|---|---|---|
| Sound Reduction Index | 17.8 | 18.7 | 20.0 | 23.3 | 21.6 | 22.1 |
| Absorption Coefficient | 0.02 | 0.02 | 0.03 | 0.04 | 0.04 | 0.03 |

The sound source was placed in the corridor in three different positions: in front of the bedroom door, shifted laterally by 2.5 m and by 5 m. The sound power level of the

source was set equal to the omnidirectional source present in the data base of the software; the calibration of the sound power level of this source was not considered necessary, since results of simulations are evaluated only to compare two different configurations of the bedrooms.

A subdivision level $8 \times 2^{10}$ was used for the calculation. The rays were followed for a duration of 2 s, with a time resolution of 0.01 s, and the diffraction level on the obstructing surfaces was set to 2 to keep count also of double diffraction over the obstacles between source and receivers.

The environment was assumed to have 50% humidity and 20 °C of temperature.

## 4. Results

### 4.1. Acoustic Measurements Results

Results of the standardized level difference, $D_{nT}$, between the two bedrooms and between the corridor and the bedroom, with the doors closed, are shown in the frequency range 100–5000 Hz in Figure 5.

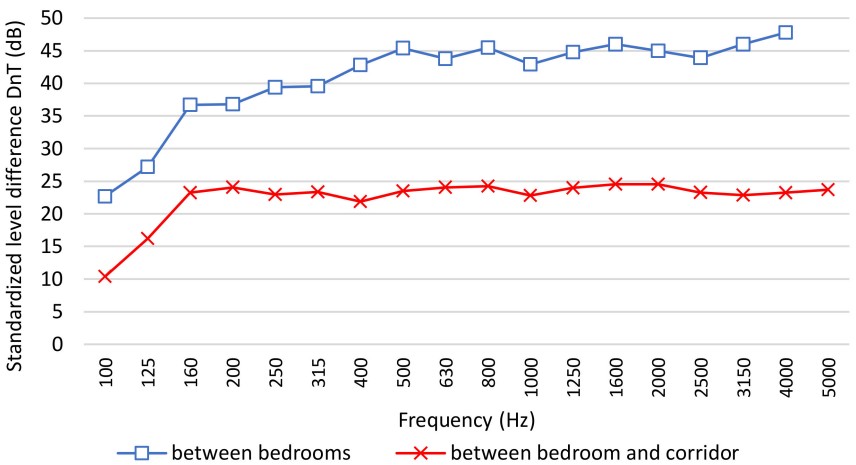

**Figure 5.** Standardized level difference, $D_{nT}$, between the two bedrooms and between the corridor and the bedroom, with the doors closed ($D_{nT,w}$ between bedrooms = 44 dB; $D_{nT,w}$ between bedroom and corridor = 24 dB).

The results shown in Figure 5 clearly highlight that the insulation between bedrooms is significantly higher than the insulation between corridor and bedroom. Under these conditions, it is possible that the insulation between bedrooms can be influenced by the transmission path through the corridor especially at higher frequencies, where the difference between the values is greater, as described in Appendix H of ISO 12354-1 [34].

Figure 6 shows the standardized level difference between the room and the corridor in the frequency range 100–5000 Hz in the configuration with the door closed, half-open, and completely open.

The single number values and the spectrum adaptation terms ($C$; $C_{tr}$) of airborne sound insulation between the bedrooms and between the bedroom and the corridor in the three configurations are shown in Table 3.

In the case of the partition between bedroom and corridor, values of Sound Reduction Index, $R'$, are not shown, since the transmission is determined mainly by the door leakages; therefore, it would not be correct to normalize the sound insulation between the adjoining rooms to the surface of the whole partition to obtain $R'$.

The second analysis carried out to study the noise immission in an unoccupied bedroom consisted of the continuous measurement of the sound levels for two days and nights, as shown in Figure 7.

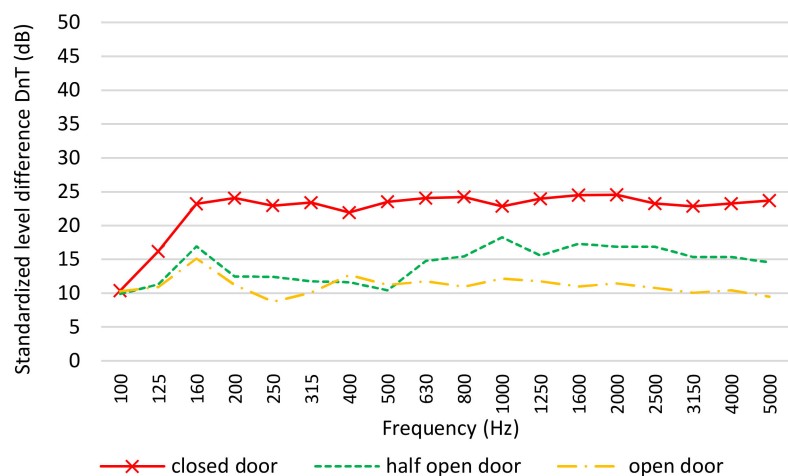

**Figure 6.** Standardized level difference between the room and the corridor in the configuration with the door closed, half-open, and completely open ($D_{nT,w}$ with closed door = 24 dB; $D_{nT,w}$ with half-open door = 15 dB; $D_{nT,w}$ with open door = 11 dB).

**Table 3.** Single number values and spectrum adaptation terms of airborne sound insulation between the bedrooms and between the room and the corridor in the three configurations.

| Airborne Sound Insulation | $D_{n,w}$ (dB) | $D_{nT,w}$ (dB) | $R'_w$ (dB) |
|---|---|---|---|
| Between bedrooms | 41 (dB) (−1.4; −5.2) | 44 (dB) (−1; −4.8) | 44 (dB) (−1.3; −5.1) |
| Between corridor and bedroom (Closed door) | 20 (dB) (0.1; −0.7) | 24 (dB) (−0.5; −0.3) | not meas. |
| Between corridor and bedroom (Half-open door) | 12 (dB) (−0.2; −0.8) | 15 (dB) (0.2; −0.4) | not meas. |
| Between corridor and bedroom (Open door) | 8 (dB) (−0.3; −0.1) | 11 (dB) (0.1; 0.3) | not meas. |

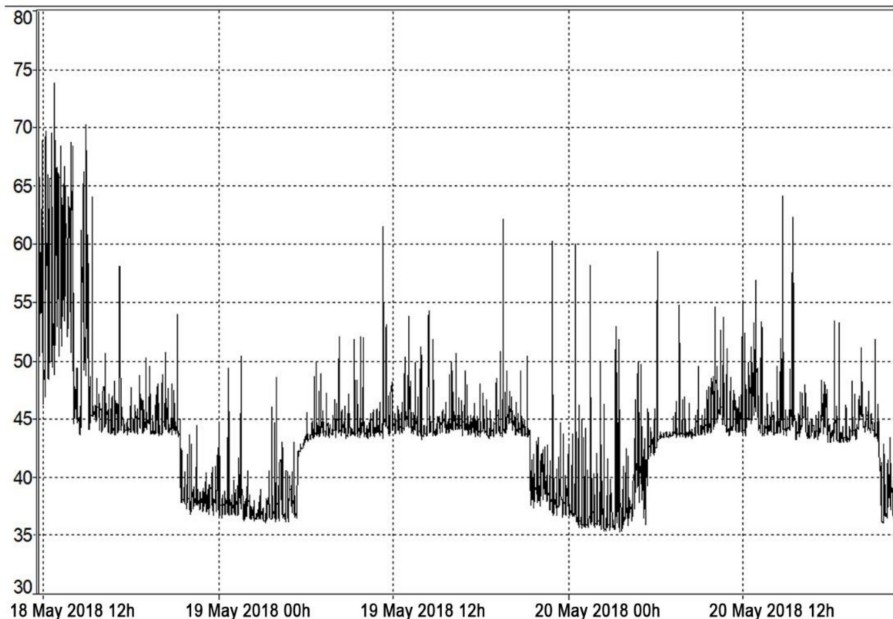

**Figure 7.** Time history of the whole sound pressure level measurement carried out in an unoccupied bedroom.

Figures 8 and 9 show the time history of the two-night periods with the exceedances of the 45 dB(A) threshold highlighted in green.

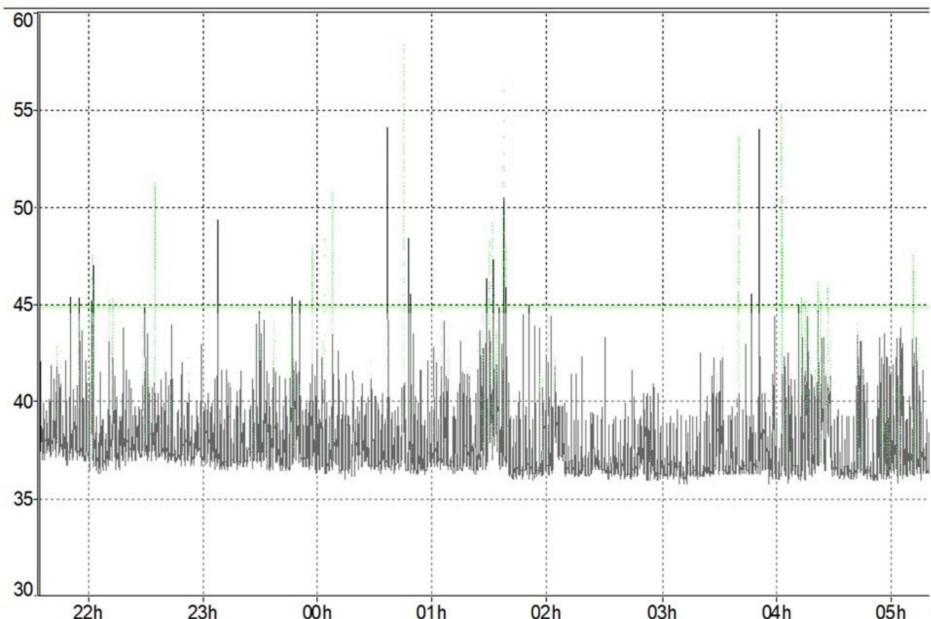

**Figure 8.** Time history of the first night (in green, the threshold exceedances set to $L_{Aeq}$ = 45 dB(A)).

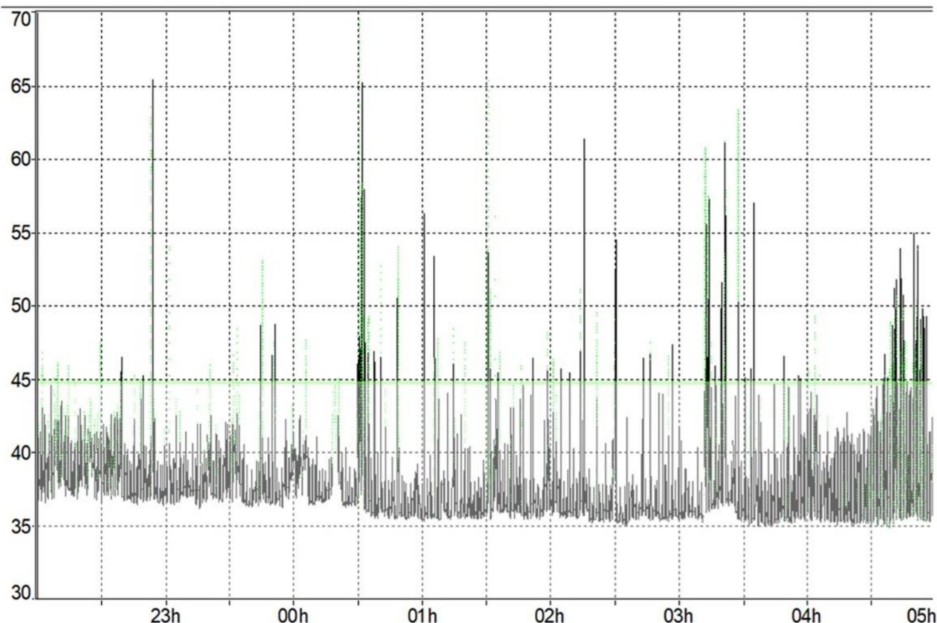

**Figure 9.** Time history of the second night (in green, the threshold exceedances set to $L_{Aeq}$ = 45 dB(A)).

It is evident that during the two nights of measurement, numerous events occurred that exceeded the level set as the wake-up threshold (45 dB(A)).

The number of total events with $L_{Aeq}$ > 45 dB(A) during the first night is 50 while the second night period is characterized by a much larger number of events, 191, quadrupling those recorded the previous night, as shown in Figure 10.

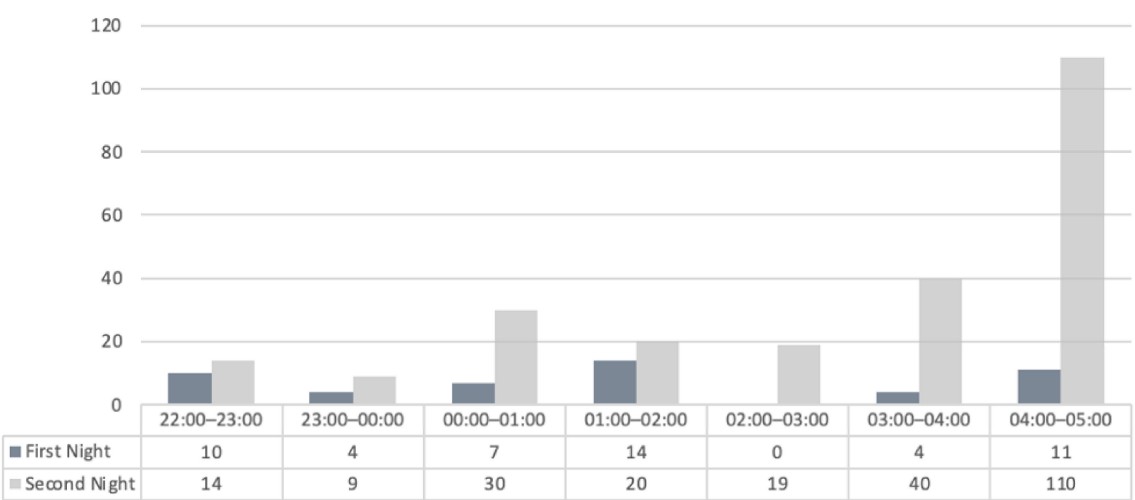

**Figure 10.** Number of events with $L_{Aeq} > 45$ dB(A) during the first and second night.

*4.2. Survey Results*

According to the results of the questionnaires, it was found that 65% of the patients are awakened at night by noise (often 27% and sometimes 38%, in Figure 11) [1].

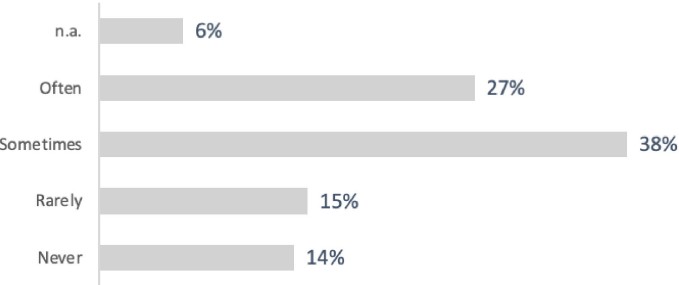

**Figure 11.** Results of the survey: "How many times did you wake up at night for noise?".

Figure 12 shows in detail the exact cause of noise from inside (above) or outside (below) the room, according to the patients interviewed.

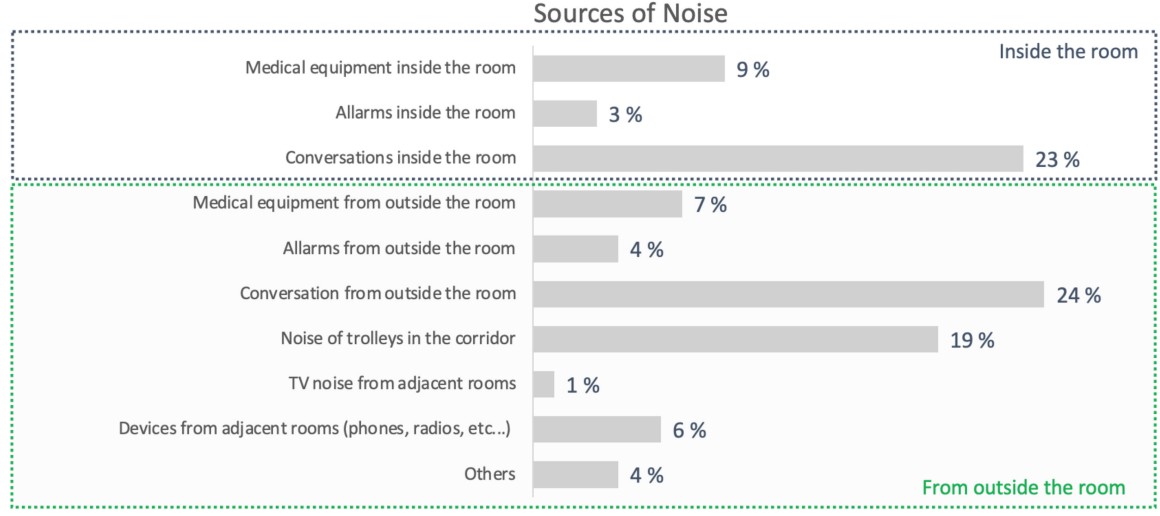

**Figure 12.** Results of the survey: "What sources of noise mainly woke you up during the night?

It is evident that patients are disturbed during the nighttime especially by conversation within the room (23%), by conversation from outside the room (24%), and by the passage of the carts in the corridor (19%).

### 4.3. Results of Simulations

Measurements and surveys carried out lead to the consideration that sound transmission between corridor and bedrooms is particularly critical. This transmission largely depends on the insulating performance of the door. However, the room entrance design and its side surfaces finishing (walls and ceiling) are very important too.

To study the effect of these aspects, simulations of two alternative configurations of the same room were carried out by using the Ramsete® ray tracing simulation program.

The characteristics of the partitions and of the door between the bedroom and the corridor were kept the same in the two simulations and are described in Section 3.4.

To limit reverberation in the room access space, sound absorbing wooden grooved acoustic panels were applied to the walls (surface mount) and to the ceiling (cavity mount). These panels were chosen for their easily cleanable surfaces and high absorption properties at medium-high frequencies; the absorption coefficients of the panels, for both kinds of installation, are shown in Table 4.

**Table 4.** Absorption coefficient of sound-absorbing wooden grooved acoustic panels applied to the walls (surface mount) and to the ceiling (cavity mount) of the room access space.

| Frequency (Hz) | 125 | 250 | 500 | 1000 | 2000 | 4000 |
|---|---|---|---|---|---|---|
| Abs. coeff. cavity mount | 0.15 | 0.40 | 0.90 | 1.00 | 1.00 | 0.90 |
| Abs. coeff. surface mount | 0.50 | 0.88 | 1.00 | 1.00 | 1.00 | 0.98 |

To maximize the sound-absorbing effect, linings were applied to the side walls and to the ceiling of the room access space; the shape of this space was modified as shown in Figure 13. In this configuration, the sound propagating from the corridor to the bedroom is partly absorbed during its reflection on the surfaces of this access space.

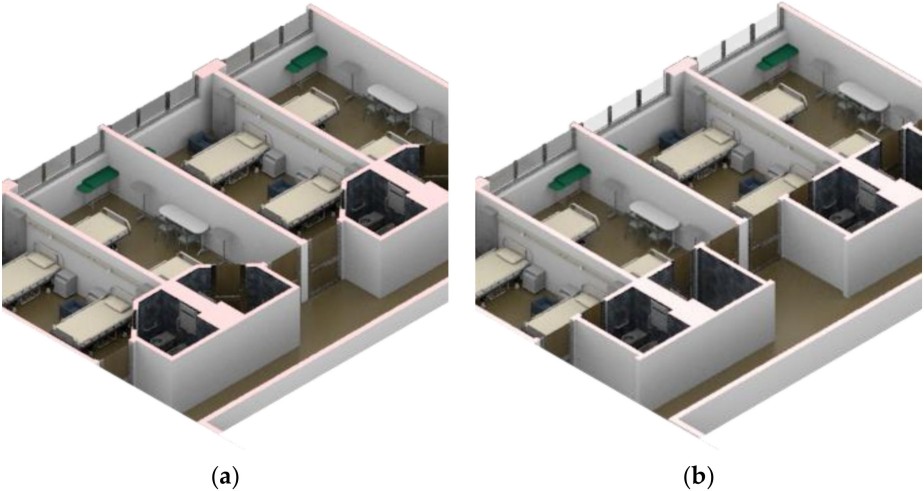

(**a**)                          (**b**)

**Figure 13.** The two alternatives of access space to the bedrooms: on the left, in the original configuration (**a**) and on the right, in the modified configuration, with the different shape of the room access space (**b**).

Simulations were carried out by placing a sound source in different positions in the corridor, as described in Section 3.4.

Figure 14 shows the results with the position of the source 5 m shifted laterally in the corridor.

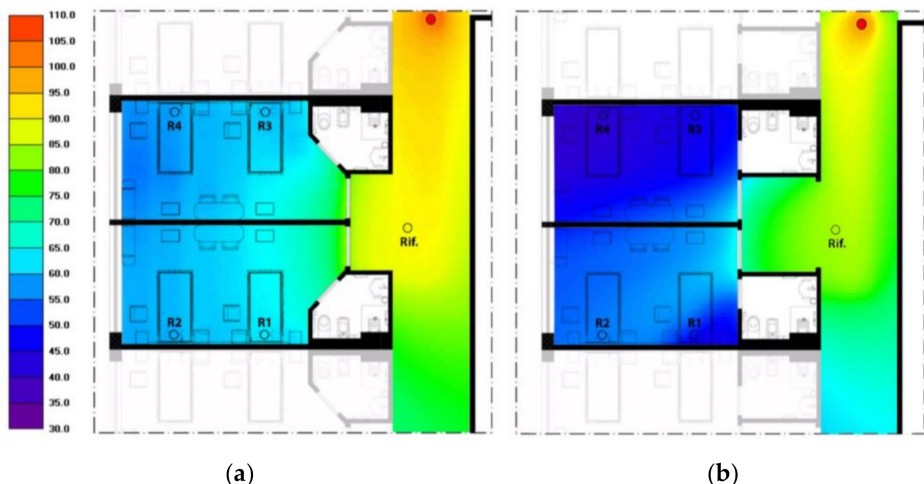

**Figure 14.** Results of simulations of sound propagation generated by a source in the corridor: on the left, in the original configuration (**a**) and on the right, in the modified configuration (**b**). Adapted with permission from Ref [36].

Thanks to the bedroom access space modification and the application of sound-absorbing materials to its lateral surfaces, a considerable improvement is achieved inside, by reducing sound pressure level by about 11 dB.

## 5. Discussion and Conclusions

The airborne sound insulation measurements in the new maternity ward of the Florence Hospital of Careggi confirmed that, within a typical inpatient room, the weak element, in terms of acoustic performance, is the partition between the bedroom and the corridor and, in particular, the door.

The measurement results showed that the sound insulation between the two adjacent bedrooms ($D_{nT,w}$ = 44 dB) did not agree with the performance required by Italian acoustic requirements for hospitals ($D_{nT,w} \geq 50$ dB); this lack of insulation is probably due to the performance of the lightweight partition between the bedrooms but also to the transmission path through the corridor. Indeed, also the sound insulation between the bedroom and the corridor was found to be inadequate ($D_{nT,w}$ = 24 dB) compared to the requirements of Italian regulations ($D_{nT,w} \geq 30$ dB). When considering the more usual position of the door (half-open or completely open), the sound insulation is further reduced to 15 dB (half-open) and to 11 dB (completely open).

The time history of the Sound Pressure Level monitoring, carried out in an unoccupied closed room, led to the conclusion that the number of sound events capable of waking up residents is very high and distributed throughout the night.

As expected, the responses to the questionnaires confirmed that most of the disturbing noise perceived inside the bedroom seems to come from outside the room (65%). Specifically, most of the noise from the corridor was due to hospital staff activities, such as conversations (24%) and carriage movement (19%). Seven percent of the perceived noises seems to come from adjacent rooms, most likely due to the sound transmission between the two rooms through the room–corridor–room path. Alarms, whether they are inside (3%) or outside (4%) the room, were not found to be a relevant disturbing source within this hospital ward during night hours. A significant noise source (23%) was also found to be due to conversations within the patients' rooms.

The results of the sound insulation measurements, of the acoustic monitoring and of the questionnaires confirmed that most of the disturbing noises within the maternity ward are generated by users' behaviors as medical staff, patients, and visitors, in the corridor and inside the room.

According to the measurements results and to the simulations, the door between the bedroom and the corridor is the acoustically weakest element; moreover, it was found

that this door is almost always kept open or half-open by the sanitary staff, for safety reasons, even at night. For this reason, we can consider that its replacement with a door characterized by a better sound reduction index may not be the most appropriate solution.

A greater sensitivity of the staff to noise disturbance, limiting staff activities in the vicinity of the patient rooms, could help to keep a more respectful and quieter environment, not causing disturbance to patients during the night hours.

The measurements and simulations showed a significant variation in the sound level within the bedroom, showing more exposed and more masked points. This is due to the bedroom and access space spatial configuration.

Simulations carried out show that a better bedroom access space layout reduces the noise propagation coming from the corridor toward the bedroom, also without modifying the characteristics of the door.

The continuation of the study will concern the research for the best configuration for the access space between the bedroom and the corridor according to a design layout, which, while preserving the necessary requirements of hospital functionality (single-access door), would reduce sound propagation between the corridor and the bedroom.

**Author Contributions:** Conceptualization: V.A., L.M., N.S. and S.S.; methodology, measurements, and simulations: V.A. and S.S. All authors have read and agreed to the published version of the manuscript.

**Funding:** This research received no external funding.

**Data Availability Statement:** This paper is an extended version of the paper presented at the Inter Noise Conference 2019 [1].

**Acknowledgments:** The authors would like to thank Gianfranco Cellai and architect Riccardo Panichi, from the University of Florence, and architect Giacomo Bai, from the University of Pisa, for their support in carrying out the measurements and the simulations. They would also like to thank the medical staff of the Careggi hospital for the availability of the spaces where the measurements were taken.

**Conflicts of Interest:** The authors declare no conflict of interest.

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
