# Peer review of "Analysis of the Acoustic Comfort in Hospital: The Case of Maternity Rooms"

_buildings, doi:10.3390/buildings12081117_

Round 1

Reviewer 1 Report

Dear authors,

The research presents in this paper a very interesting topic, as well as results that are of wider significance when it comes to the improvement of noise characteristics of hospitals.

1.       The abstract is clear and gives a good indication of what the article is focusing on.

2.       The introduction is very well written with a good connection to references.

3.       The methods are well described and data for measurement equipment is given. No additional charges are requested at this point.

4.       The results are presented clearly with no methodical errors detected. However Fig 7-9 are problematic to read due to being a print screen from the software. If possible please exchange them with graphs where axes and values are readable. Also please enlarge Fig.10 so the full overview is visible.

4.       The results discussion is clear.

7.       The biggest problem of the article is a weak statement of novelty. Some findings are quite obvious and without a stronger statement o novelty, the originality of the study is limited.

Editing comments:

1.    When referencing more than two bibliography sources in order like [3,4,5] it is customary to write it as [3-5]. Please check the whole article.

2.    Additionally, please avoid using conditional forms like “can”, “could”, and “should”. If possible strong statements must be used in the scientific paper. Otherwise, it seems that the author is not sure about some of the thesis mentioned.

Conclusions:

The article although interesting at parts requires some adjusting of text (in case of novelty statement) and editing. The quality of some of the figures must be improved.  

Author Response

Dear reviewer, thank you very much for your suggestions and corrections to our manuscript, which improved its quality.

We have revised the text and many images in accordance with your comments and those of the other reviewers.

Best regards.

Reviewer 2 Report

The manuscript entitled “Analysis of the acoustic comfort in hospital: the case of maternity rooms” conducted field measurements and simulations in order to analyse the acoustic quality of maternity rooms. I have several comments or questions that need to be addressed before this paper could be accepted for publication.

 1)     Abstract section exceeds the words limit (> 200).

2)     Manuscript structure should be revised. There are many one-sentence paragraphs.

3)     “There may be many causes of stress, but environmental discomfort is usually related to noise disturbance and inadequate lighting.” Please, add reference.

4)     There are no units in the colour scales in Figure 15. Use the same colour scale in both figures to make it easier to compare the obtained results.

5)     “In order to make data of different nature interoperate, the building was modeled with a BIM-oriented software. BIM modeling allowed us, in addition to defining the three dimensional model of the building, to attribute the material characteristics of the building elements that define the patient room.” Authors should specify the nature of the different data and the material characteristics add to the BIM model. Please add a Table with the parameters that need to be defined and indicate the attribute name, description, entity (material, room, area, wall, etc.), data type (text, integer, etc.), etc.

6)     Since the simulation process was conducted using Ramsete software, what was the purpose of generating the BIM model?

7)     3.3 Questionnaires section. “Questions were aimed at estimating the acoustic quality of the nighttime hours within the department and what the most disturbing noise sources were.” How many questions were included in the questionnaire to estimate the acoustic quality? Authors should include these questions in this section.

8)     Does the questionnaire include questions about participant’s characteristics (e.g. gender, age, number of nights spent in hospital, etc.)?

9)     Authors should add a “Conclusion” section with the most important conclusions and take-home messages.

Author Response

Dear reviewer, thank you very much for your suggestions and corrections to our manuscript, which improved its quality.

We have revised the text and many images in accordance with your comments and those of the other reviewers.

Reviewer 3 Report

 The case study is relevant and actual. The measurement methods were presumably adequate but the paper would benefit from more detailed description of the instrumentation and methods applied in situ. The description of the room acoustical modeling is not sufficient. Too many details on initial data, applied methods and reasoning on selected application are missing. The results of questionnaires are supported by the measured data. Therefore it is not sure if the modeling of the rooms is providing added value in this study. The room acoustical simulation of hospital environment could be a topic in another paper.

Many references are Italian publications. It would be beneficial if similar (research) publications were in English because the intended readers of this publication are not supposed to comprehend Italian.

There are no conclusions in this study?

There are several little language mistakes that should be corrected. The text could benefit from a review by native English speaking researcher. 

 Some detailed comments:

Introduction

2nd par, several studies show etc., This requires more than one reference to convince the statement.

4th par, For healthcare staff, can alter stress level, reducing work performance etc. Does it mean that stress level reduces work performance and increases burnout rates or that prolonged exposure to noise can alter, reduce, and increase?

p.2, 4th par, “excessive noise pollution levels”, recommended phrase is “excessive noise levels”

same chapter what is the meaning of gravedance?

p.2, 6th par, exceed this recommended range, change “range” to “value” or “limit” since it is single number value 50 dB(A). dB(A) should be used consistently in the text.

p.2, 7th par, “In line with numerous scientific studies showing negative relationships…,” The statement should be supported by references to such studies.

p.2, 2nd bullet “continuous measurements of sound pressure level” add “long-term” between continuous and measurements

p2, 3rd bullet could be questionnaires distributed to the patients

p2, the last paragraph is repetition of the previous bullet points.

Acoustic requirements for hospital bedrooms

p3, end of 3rd par, applicable only to public hospital..., maybe use plural “hospitals”

p3, 4th par, refer to the Italian standards. There is one reference for sound environment requirements (sound insulation, impact noise and noise from equipment)” and another for room acoustic requirements. “Acoustic insulation performance” should be “Sound insulation performance”.

p3, just before Table 1, “Decree 5/12/1997”, please use same citation as above Decree of December 5,1997 [26].

Materials and Methods

p4, 2nd par, “the ground surface”, perhaps could use “the floor area”

p4, 3rd par, the text is difficult to follow. Please simplify the sentences.

p5, Figure 1 caption, “system of the maternity ward of the” the sentence is not completed. Please rephrase.

p5, 1st par, BIM acronym should be defined. Is this information relevant for the study?

p5, 2nd par, “…between the door and the lateral wall and the paving” The paving reminds of street and sidewalk. In this case maybe a carpet would be more suitable description of the floor covering. The last sentence is missing a point.

p5, sec 3.2., 1st par, it should cite to correct part of standard ISO 16283-1.

p5, sec 3.2, 2nd par, instrument types could be described in more detail: e.g., microphone manufacturer, type, model. Same for loudspeaker, calibrator etc.

p6, after Fig. 4, Continuous long-term sound pressure level, SPL, measurements. The analyzer type, model etc. could be described.

p7, sec 3.3, 2nd par, both to the acoustic, lighting and colorimetric issues. There are more than two issues. Please clarify the whole paragraph.

p7, sec 3.4. 1st par, citation to the method reference is necessary.

p8, Fig.5, DnT,w values should be presented to enable comparison to requirements of Table 1.

p8, 2nd par, preferably use phrase “the sound insulation between bedrooms is significantly higher than the sound insulation between corridor and bedroom”. Try to avoid using “twice” with dB-results since it is logarithmic scale.

p8, Fig 6 DnT,w values should be presented to enable comparison to requirements of Table 1. The Figs 5&6 should be in the same format to enable direct comparison.

p9, Table 3, R´w apostrophe is in subindex, please correct.

p9, Figs 7 & 8, The axis labels are with too small font to read.

p10, Fig9, The axis labels are with too small font to read. The threshold could be shown in the figure. The threshold should also be explained in more detail in methods section. Presumably it is A-weighted SPL using fast-time constant but it might also be slow-time constant.

p10, Fig 10 presents events with LAmax>45 dBA but in the text above the number of total events with LAeq>45 dBA is referred. Therefore, it is vague which is the sound level parameter that has been studied.

p11, Table 4, this should be in the methods not in the results. Actually, all the chapters 4.3. on page 11 are describing methods. The simulation methodology should be described in more detail. For example, how were the models calibrated? What considerations were made to ensure that modeled results are reasonable?

p12, Fig 13, the changes in configurations could be illustrated more clearly. Fig 14 could be combined with Fig 13.

p12, Fig 15 could show the level difference between the configurations inside the room since it is the result that the reader is expecting.

p13, 2nd par, why is the reverberation time inside the bedroom affected by the changes in the corridor’s room acoustics?

p13, 5 Discussion, 3rd par, “acoustic monitoring” was previously described as time history SPL measurements. The acoustic monitoring should be introduced in methods.  

p13, 5 Discussion, 5th par, “perceived noises cames...” presumably “…comes…”

References

6 the citation is not complete. Please add publisher information.

11 the citation is not complete. Please add publication information.

24 in in the process of … remove first in

32 Applied Acoustics should be Applied Acoustics 1997,

35 Neo-Eubios 71

Author Response

(The authors gave the same response as above.)

Round 2

Reviewer 2 Report

The Authors have augmented the quality of the manuscript sufficiently.